# Harmonizing MR Images Across 100+ Scanners: Multi-site Validation with Traveling Subjects and Real-world Protocols

**Savannah P. Hays**[1] [ID]                                            SHAYS6@JHU.EDU
[1] *Image Analysis and Communications Laboratory, Department of Electrical and Computer Engineering, Johns Hopkins University, USA*

**Lianrui Zuo**[2] [ID]                                            LIANRUI.ZUO@VANDERBILT.EDU
[2] *Department of Electrical and Computer Engineering, Vanderbilt University, USA*

**Muhammad Faizyab Ali Chaudhary**[1] [ID]                           MCHAUD25@JHU.EDU
**Kathleen M. Bartz**[1] [ID]                                       KBARTZ2@JHU.EDU
**Samuel W. Remedios**[1] [ID]                                     SREMEDI1@JHU.EDU
**Jinwei Zhang**[1] [ID]                                            JWZHANG@JHU.EDU
**Jiachen Zhuo**[3] [ID]                                      JZHUO@SOM.UMARYLAND.EDU
[3] *Department of Diagnostic Radiology and Nuclear Medicine, University of Maryland School of Medicine, USA*

**Murat Bilgel**[4] [ID]                                       MURAT.BILGEL@NIH.GOV
[4] *Laboratory of Behavioral Neuroscience, National Institute on Aging, USA*

**Shiv Saidha**[5] [ID]                                            SSAIDHA2@JHMI.EDU
[5] *Department of Neurology, Johns Hopkins School of Medicine, USA*

**Ellen M. Mowry**[5] [ID]                                         EMOWRY1@JHMI.EDU
**Scott D. Newsome**[5] [ID]                                       SNEWSOM2@JHMI.EDU
**Jerry L. Prince**[1] [ID]                                          PRINCE@JHU.EDU
**Blake E. Dewey**[5] [ID]                                     BLAKE.DEWEY@JHU.EDU
**Aaron Carass**[1] [ID]                                      AARON_CARASS@JHU.EDU

**Editors:** Accepted for publication at MIDL 2026

## Abstract

Reliable harmonization of heterogeneous magnetic resonance (MR) image datasets, especially those acquired in pragmatic clinical trials, is critical to advance multi-center neuroimaging studies and translational machine learning in healthcare. We present an enhanced and rigorously validated version of the HACA3 harmonization algorithm, which we refer to as HACA3$^+$, incorporating key methodological enhancements: (1) an improved artifact encoder to better isolate and mitigate image artifacts, (2) background and foreground-sensitive attention mechanisms to increase harmonization specificity, and (3) extensive training using data spanning 100+ scanners from 64 independent sites, providing a broader diversity of scanners than other harmonization methods. Our study focuses on four commonly acquired MR image contrasts (T1-weighted, T2-weighted, proton density, & fluid-attenuated inversion recovery), reflecting realistic clinical protocols. We perform inter-site harmonization experiments using traveling subjects to assess the generalization and robustness of the harmonization model. We compare the results of the publicly available version of HACA3 and our implementation, HACA3$^+$. Downstream relevance is further established through whole brain segmentation and image imputation. Finally, we justify each enhancement through an ablation experiment. Pre-trained weights and code for HACA3$^+$ are made publicly available at https://github.com/shays15/haca3-plus.

**Keywords:** MRI, Image Harmonization, Image Synthesis

## 1. Introduction

Variability in magnetic resonance (MR) image acquisition due to different scanner parameters and manufacturers leads to inconsistent image contrasts and intensities across datasets, which can severely confound biomarker development and analysis. Multi-site neuroimaging studies and the adoption of deep learning in clinical MR image processing workflows depend critically on the ability to harmonize MR imaging data to overcome this variability. However, despite advances in algorithm development, few harmonization methods undergo rigorous, large-scale validation with multiple out-of-domain datasets, which limits the algorithms translational impact while also undermining reliability.

A broad spectrum of harmonization methods for neuroimaging have been proposed, which can be broadly grouped into statistical and image-based approaches. Statistical harmonization methods, such as neuroComBat (Fortin et al., 2018) and its extensions (Beer et al., 2020; Horng et al., 2022), operate on derived measurements (e.g., cortical thickness and diffusion metrics) to adjust for scanner or site effects while preserving biological variability and have been widely applied (Fortin et al., 2018; Horng et al., 2022). In contrast, image-based harmonization aims to correct scanner-dependent variation directly at the voxel level using generative models. This enables harmonization before downstream analyzes and allows a single corrected image to support multiple tasks (Dewey et al., 2019, 2020; Liu et al., 2021). Although image-based methods offer greater flexibility, they often require substantially larger training data and may be sensitive to domain shifts.

Earlier image-based harmonization methods, such as DeepHarmony (Dewey et al., 2019), used supervised training on paired images, requiring the same subject to be scanned across different scanners and possibly sites. However, reliance on paired data limits their broader applicability as (inter-site) traveling subject data is almost never available. To relax the assumption of paired data, unpaired image-to-image translation using cycle-consistency was proposed (Modanwal et al., 2020). Although these models use distribution matching for synthesizing samples across domains, they are limited in two pertinent ways: their inability to preserve anatomical structure across domains and limited scalability across sites and contrasts. To address this, a major shift was observed towards anatomy-contrast disentanglement approaches (Liu et al., 2017; Huang et al., 2018).

Recent methods such as HACA3 (Zuo et al., 2023) and MURD (Liu and Yap, 2024) have introduced attention mechanisms, flexible contrast handling, and unified multi-domain architectures. HACA3 relaxes assumptions about contrast similarity and incorporates artifact-aware modules to robustly harmonize images across diverse contrasts and protocols, even in the presence of motion or noise. Meanwhile, MURD addresses scalability by learning a single encoder-decoder framework that generalizes across sites without direct supervision, enabling realistic harmonization while preserving subject-specific features.

Recent work by Lu et al. (2025) demonstrated that harmonization performance can vary substantially between scanners and sequences, motivating the critical importance of robust, large-scale validation of MR image harmonization methods to ensure their effectiveness and reliability for subsequent analyzes. Although all previously reported harmonization methods have shown a benefit over using unharmonized data, HACA3 (Zuo et al., 2023) showed the most consistent results in comparison to DeepHarmony (Dewey et al., 2019) and neuroCombat (Fortin et al., 2018) in Lu et al. (2025). Another recent work by Ho et al.

([2026](#)) further compared neuroCombat and HACA3 showing that HACA3 outperformed neuroCombat. Supervised methods that use paired training data may outperform HACA3, but when using real-world, large-scale datasets, HACA3's ability to handle unpaired data and missing contrasts gives it a practical edge.

HACA3[1] is an unsupervised image harmonization algorithm for structural MR neuroimages, which does not require paired subject data for training. The HACA3 model is built on an encoder-decoder structure. Three separate encoders disentangle representations of anatomical structure, acquisition contrast, and image quality (artifact level) from each input MR image. The network employs a contrastive learning framework to ensure that the extracted anatomical latent space remains domain-invariant, while maintaining distinct representations for image contrast and artifact. Harmonization is achieved by recombining anatomy and contrast encodings to synthesize images with desired contrast and quality characteristics. This architectural flexibility allows HACA3 to handle unpaired or incomplete contrast data, impute missing contrasts, and operate unsupervised across diverse sources.

In this work, we focus on enhancements to HACA3 that address three of its limitations. First, we expand on work by Hays et al. ([2025b](#)) to include a margin loss in the contrastive learning framework to score an MR image's artifact level. We use a 2D score to replace the HACA3 artifact encoder and train this new artifact encoder on a larger range of simulated artifacts. The margin loss within the encoder allows for the stratification by different artifact levels, which helps improve its sensitivity.

Our second enhancement to HACA3 introduces an attention mechanism that operates on a 2D slice-wise level, in all three cardinal orientations, covering the volume. HACA3's current scalar attention vector is constant over an entire 2D slice regardless of the spatial location within the slice. When MR images are acquired with a full field-of-view (FOV), this does not raise concerns. However, when images are acquired with a limited FOV, such as the axial plane missing a portion of the superior skull, HACA3 will fail to recover the missing region even if it was present in other input source images. Recent preliminary work ([Hays et al., 2025a](#)) focused on this modification of HACA3's attention mechanism to be background and foreground aware, allowing for variation across a 2D slice.

Our last enhancement to HACA3 is the inclusion of considerably more data in the training of the contrast encoder. HACA3 was originally trained on MR images from 21 different sites consisting of only 21 scanners, with each site contributing ten subjects. Our contribution is to train across 64 different sites covering 132 scanners, with a total of 996 subjects; this represents a six-fold increase in the number of scanners and a quadrupling in the number of subjects. We refer to our new version of HACA3, with the outlined three enhancements, as HACA3$^+$. Although many harmonization frameworks have been developed, large-scale experimental validation using traveling-subject data and diverse scanner sources remains rare. The primary aim of this work is to rigorously assess HACA3 and its enhanced variant, HACA3$^+$, on datasets spanning multiple sites, scanners, and subject populations. Unlike benchmarking studies proposing new architectures, our focus is on confirming robustness, generalization, and clinical utility at scale. Prior independent studies ([Lu et al., 2025](#); [Ho et al., 2026](#)) have demonstrated that HACA3 performs favorably to other harmonization methods under real-world settings and datasets. Recent works

---

1. [https://github.com/lianruizuo/haca3](https://github.com/lianruizuo/haca3)

([Beizaee et al., 2025](); [Lee et al., 2025]()) primarily focus on narrow modality ranges (e.g., T1w and T2w only), whereas our evaluations cover a broader set of domains. Accordingly, our experiments center on HACA3 and HACA3$^+$ rather than extensive baseline methods, and our contribution lies in unprecedented scale and clinical relevance. This paper provides a comprehensive, clinically-relevant validation of the HACA3$^+$ algorithm for MR image harmonization. Throughout the paper, we compare HACA3$^+$ with the publicly available version of HACA3. The pre-trained weights and code for HACA3$^+$ are made publicly available at https://github.com/shays15/haca3-plus.

## 2. Methods

### 2.1. Technical Contributions of HACA3$^+$

**Enhanced Artifact Encoder**    The first technical contribution is the modification of the artifact encoder. We trained the artifact encoder using 297 structural MR volumes from the TRaditional vs. Early Aggressive Therapy for Multiple Sclerosis (TREAT-MS) pragmatic, clinical trial (NCT03500328) ([Mowry et al., 2025]()). These scans were acquired from seven different imaging sites and included four structural MR image contrasts: $T_1$-weighted ($T_1$-w), $T_2$-weighted ($T_2$-w), fluid-attenuated inversion recovery (FLAIR), and proton density (PD) images. Only high-quality images were included in the training dataset. Prior to training, the images were N4 bias field corrected ([Tustison et al., 2010]()) and 2D acquisitions were super-resolved ([Remedios et al., 2023]()). Similarly to HACA3, we simulated common MR image artifacts using the TorchIO library ([Pérez-García et al., 2021]()), including random noise, random ghosting, random bias field, and random anisotropy. Unlike HACA3, we mapped the artifact simulation parameters to a normalized score, where a low artifact level mapped to a low score and a high artifact level mapped to a higher score. We incorporated this score into training through the triplet loss. The triplet loss enforces a ranking such that the clean anchor image receives a lower severity score than the artifact-degraded negative image:

$$\mathcal{L}_{\text{triplet}} = \sum_{i=1}^{N} \max(0, S_i^{\text{anchor}} - S_i^{\text{positive}} + m) + \max(0, S_i^{\text{negative}} - S_i^{\text{anchor}} + m) \qquad (1)$$

where $m$ is a dynamic margin based on artifact severity.

**Enhanced Attention**    The second technical contribution is the modification of the attention module to handle limited FOV source images. HACA3 uses the background mask of only the first input source image. This mask is used for all synthetic images independent of the harmonization target and attention for each source image. Our modification uses the union of the background mask from all of the source images. This modification only makes a difference if there are multiple source images. For each pixel within the union background region, the attention will be equally distributed across source images. For each pixel within the union foreground region, the attention will be distributed across source images according to the similarity between the source image and the target image as computed in HACA3. For other pixels not in the union regions, the background source pixels will be forced to 0 and the attention between the foreground source pixels, as determined by the attention module, will be normalized to sum to 1.

Our approach for limited FOV imputation relies on multiple contrast source images to accurately impute missing regions. Imputation only occurs when a region is visible in at least one source image. We chose not to impute regions without any information across the source images. To validate this modification, we simulated limited FOV images using MR images (N=61) from 7 sites included in Table 1. Limited FOV images were simulated by cropping regions from full FOV images. We simulated two types of limited FOV acquisitions: anterior degradation and left/right degradation. We tested HACA3$^+$ and HACA3 using simulated degradations on $T_1$-w, $T_2$-w, and FLAIR images. Evaluation metrics include peak-signal-to-noise ratio (PSNR) and structural similarity index measure (SSIM) between the imputed, harmonized image and the acquired full FOV image.

## 3. Data: Training and Validation

### 3.1. Training Dataset

The training datasets used for HACA3$^+$ are from 64 imaging sites and 132 scanners. These datasets are summarized in Table 1. The four MR image contrasts used in training are: $T_1$-weighted ($T_1$-w), $T_2$-weighted ($T_2$-w), fluid-attenuated inversion recovery (FLAIR), and proton density (PD). Not every site acquired all four contrasts, which is indicated in Table 1. All subjects included in the training have at least a $T_1$-w image. In total, 996 subjects were used to train HACA3$^+$. The open sites were from the following datasets: Open Access Series of Imaging Studies-3 (OASIS-3) (LaMontagne et al., 2019), Baltimore Longitudinal Study of Aging (BLSA) (Resnick et al., 2000), Human Connectome Project (HCP)[2], and Information eXtraction from Images (IXI)[3]. We included data from multiple imaging sites within the OASIS-3, BLSA, and IXI datasets. These are labeled as different sites in Table 1. In total, there were 11 open sites used in training. The private data included 46 sites from the TREAT-MS pragmatic clinical trial (Mowry et al., 2025). The remaining seven sites are from other private sources.

For preprocessing, all images were background removed followed by N4 bias field correction (Tustison et al., 2010). Super-resolution (Remedios et al., 2023) was applied to 2D acquisitions. Finally, 3D acquisitions and super-resolved 2D acquisitions were rigidly registered to the MNI152 atlas using ANTs (Avants et al., 2009). The 50 middle slices of each orientation were extracted from each MR image volume for training, resulting in 150 slices per volume.

---

2. https://www.humanconnectome.org/study/hcp-young-adult
3. https://brain-development.org/ixi-dataset/

Table 1: Training site characteristics and MR contrast availability for HACA3$^+$. S1-S21 are the sites used to train the publicly available HACA3. Open refers to the public availability of the site data. The manufacturers (Manu.) are: G - GE, H - Hitachi, P - Philips, S- Siemens, T - Toshiba. The field (Field) strength column lists used field strengths in Telsa (T) at a site. The population (Pop.) column codes are: HC - healthy controls, MS - multiple sclerosis, TB - mild tramatic brain injury. For T1w, T2w, FLAIR, and PD, the number denotes the number of used scans for training HACA3$^+$.

| Site ID | Open | Manu. | Field (T) | Pop. | T1w | T2w | FLAIR | PD |
|---------|------|-------|-----------|------|-----|-----|-------|-----|
| S1 | ✔ | P | 1.5 | HC | 10 | 10 | 10 | 10 |
| S2 | ✔ | P | 3.0 | HC | 10 | 10 | 10 | 10 |
| S3 | ✔ | S | 3.0 | HC | 10 | 10 | 10 | 10 |
| S4 | ✔ | S | 3.0 | HC | 10 | 10 | 10 | 10 |
| S5 | ✔ | S | 3.0 | HC | 10 | 10 | 10 | 10 |
| S6 | ✔ | S | 1.5 | HC | 10 | 10 | 10 | 10 |
| S7 | ✔ | P | 1.5 | HC | 10 | 10 | 10 | 10 |
| S8 | ✔ | P | 3.0 | HC | 10 | 10 | 10 | 10 |
| S9 | ✔ | P | 3.0 | HC | 10 | 10 | 10 | 10 |
| S10 | ✔ | P | 3.0 | HC | 10 | 10 | 10 | 10 |
| S11 | ✖ | P | 3.0 | MS | 10 | 10 | 10 | 10 |
| S12 | ✖ | P | 3.0 | MS | 10 | 10 | 10 | 10 |
| S13 | ✖ | S | 3.0 | TB | 10 | 10 | 10 | 0 |
| S14 | ✖ | S | 3.0 | HC | 10 | 10 | 10 | 0 |
| S15 | ✖ | G | 3.0 | MS | 10 | 10 | 10 | 10 |
| S16 | ✖ | G | 3.0 | MS | 10 | 10 | 10 | 10 |
| S17 | ✖ | S | 1.5 | MS | 10 | 0 | 10 | 0 |
| S18 | ✖ | S | 3.0 | MS | 10 | 10 | 10 | 10 |
| S19 | ✖ | G,H,P,S,T | 1.1,1.5,3.0 | MS | 125 | 125 | 125 | 65 |
| S20 | ✖ | G,P,S | 1.5,3.0 | MS | 22 | 22 | 20 | 0 |
| S21 | ✖ | G,P,S,T | 3.0 | MS | 26 | 26 | 26 | 1 |
| S22 | ✖ | G,H,S | 1.1,1.5,3.0 | MS | 42 | 42 | 38 | 2 |
| S23 | ✖ | G,P,S | 1.5,3.0 | MS | 26 | 26 | 26 | 2 |
| S24 | ✖ | G,P,S | 1.5,3.0 | MS | 23 | 23 | 18 | 3 |
| S25 | ✖ | G,P,S,T | 1.1,1.5,3.0 | MS | 14 | 14 | 14 | 0 |
| S26 | ✖ | G,P,S | 1.5,3.0 | MS | 10 | 10 | 9 | 0 |
| S27 | ✖ | G,P,S | 1.5,3.0 | MS | 5 | 5 | 5 | 2 |
| S28 | ✖ | G,P,S | 1.5,3.0 | MS | 8 | 8 | 8 | 0 |
| S29 | ✖ | G,S | 1.5,3.0 | MS | 13 | 13 | 13 | 0 |
| S30 | ✖ | G,H,P,S | 1.1,1.5,3.0 | MS | 14 | 14 | 15 | 5 |
| S31 | ✖ | G,H,S | 1.1,1.5,3.0 | MS | 19 | 19 | 19 | 2 |
| S32 | ✖ | G,P,S | 1.5,3.0 | MS | 37 | 37 | 36 | 4 |
| S33 | ✖ | S | 1.5 | MS | 3 | 3 | 3 | 0 |
| S34 | ✖ | G,H,S | 1.1,1.5 | MS | 7 | 7 | 7 | 0 |

| Site ID | Open | Manu. | Field (T) | Pop. | T1w | T2w | FLAIR | PD |
|---------|------|-------|-----------|------|-----|-----|-------|-----|
| S35 | ✖ | G,S | 1.5,3.0 | MS | 11 | 11 | 11 | 4 |
| S36 | ✖ | G,S | 1.5,3.0 | MS | 3 | 3 | 3 | 0 |
| S37 | ✖ | G,P,S,T | 1.5,3.0 | MS | 21 | 21 | 20 | 0 |
| S38 | ✖ | G,P,S | 1.5,3.0 | MS | 23 | 23 | 21 | 3 |
| S39 | ✖ | P,S | 1.5,3.0 | MS | 3 | 3 | 3 | 0 |
| S40 | ✖ | G,S | 1.5,3.0 | MS | 19 | 19 | 16 | 10 |
| S41 | ✖ | G,P,S | 1.5,3.0 | MS | 20 | 20 | 20 | 1 |
| S42 | ✖ | S | 1.5 | MS | 14 | 14 | 14 | 0 |
| S43 | ✖ | G,P,S | 1.5,3.0 | MS | 29 | 29 | 28 | 0 |
| S44 | ✖ | G,P,S | 1.5,3.0 | MS | 6 | 6 | 6 | 0 |
| S45 | ✖ | G,P,S | 1.5,3.0 | MS | 21 | 21 | 18 | 4 |
| S46 | ✖ | G,P,S,T | 1.5,3.0 | MS | 35 | 35 | 33 | 1 |
| S47 | ✖ | G,P,S,T | 1.5,3.0 | MS | 33 | 33 | 28 | 1 |
| S48 | ✖ | G,P,S | 1.5,3.0 | MS | 12 | 12 | 12 | 1 |
| S49 | ✖ | G,P,S | 1.5,3.0 | MS | 20 | 20 | 20 | 1 |
| S50 | ✖ | G,H,P,S | 1.5,3.0 | MS | 25 | 25 | 25 | 3 |
| S51 | ✖ | G,S | 1.5,3.0 | MS | 5 | 5 | 4 | 0 |
| S52 | ✖ | G,H,S,T | 1.1,1.5,3.0 | MS | 20 | 20 | 20 | 0 |
| S53 | ✖ | G,H,S,T | 1.1,1.5,3.0 | MS | 8 | 8 | 8 | 0 |
| S54 | ✖ | G,S,T | 1.5,3.0 | MS | 12 | 12 | 11 | 1 |
| S55 | ✖ | S | 1.5 | MS | 3 | 3 | 3 | 3 |
| S56 | ✖ | G,S | 1.5,3.0 | MS | 18 | 18 | 18 | 4 |
| S57 | ✖ | G,S,T | 1.5,3.0 | MS | 24 | 24 | 24 | 1 |
| S58 | ✖ | G,S | 1.5,3.0 | MS | 13 | 13 | 13 | 1 |
| S59 | ✖ | G,P | 1.5,3.0 | MS | 5 | 5 | 5 | 1 |
| S60 | ✖ | G,P,S | 1.5,3.0 | MS | 12 | 12 | 12 | 0 |
| S61 | ✖ | G,P,S,T | 1.5,3.0 | MS | 21 | 21 | 21 | 3 |
| S62 | ✖ | G,P,S | 1.5,3.0 | MS | 7 | 7 | 7 | 1 |
| S63 | ✖ | G,S | 1.5,3.0 | MS | 5 | 5 | 4 | 0 |
| S64 | ✖ | T | 1.5 | MS | 1 | 1 | 1 | 0 |

### 3.2. Validation Using Travel Subjects

A summary of the three datasets we use to validate HACA3$^+$ are in Table 2.

**Dataset #1: TREAT-MS Traveling Subjects** For inter-site harmonization validation, we use a private traveling subjects dataset from the TREAT-MS pragmatic, clinical trial (NCT0350032) (Mowry et al., 2025). MR images were collected from 14 participants (ages 18–60), including five healthy controls with no known neurological conditions and nine people with multiple sclerosis (PwMS) with stable MS status. Scans followed a 3-month protocol: initial session at Johns Hopkins, 3–5 sessions at other eastern United States sites, and a final session at Johns Hopkins. Each session followed the same acquisition

Table 2: Summary of the Validation Datasets. We did not use data from repeated sessions for FTHP and MASiVar datasets, as not every subject had a repeated scan.

| Dataset | # Subjects | # Sites | # Sessions |
|---|---|---|---|
| #1: TREAT-MS Traveling Subjects | 15 | 9[†] | 126 |
| #2: FTHP (Opfer et al., 2023) | 1 | 116 | 116 |
| #3: MASiVar (Cai et al., 2021) | 5 | 3[*] | 19 |

[†]: Each subject only visited 3–5 sites.

[*]: Images were acquired at three different institutions across 3–4 scanners.

protocol, including a scan and re-scan procedure with $T_1$-w, $T_2$-w, FLAIR, and PD sequences. For reasons beyond our control, some subjects did not complete all scans. We used the initial scan at Johns Hopkins as the harmonization target for HACA3 and HACA3$^+$ for all harmonization tasks. For input into HACA3 and HACA3$^+$, we use the acquired $T_1$-w, $T_2$-w, and FLAIR images. The PD image was withheld to demonstrate the imputation ability of HACA3$^+$. We compute the evaluation metrics of peak-signal-to-noise ratio (PSNR) and structural similarity index measure (SSIM) for each image contrast between the harmonized image and the acquired image at the Johns Hopkins site.

**Dataset #2: Frequently Traveling Human Phantom** The Frequently Traveling Human Phantom (FTHP) (Opfer et al., 2023)[4] provides a single subject, multi-scanner validation dataset. It contains $T_1$-w MR images of a single healthy male volunteer, acquired across 116 different scanners. To ensure one scan per scanner, we excluded repeat scans acquired on the same scanner, focusing on single subject, multi-scanner validation. For harmonization, we used the $T_1$-w image from the initial scan as the harmonization target for both HACA3 and HACA3$^+$. Whole brain segmentation was performed on both the unharmonized and harmonized images using SLANT (Huo et al., 2019), and the resulting segmentation labels were aggregated into nine brain regions for analysis. The Dice similarity coefficient (DSC) was calculated for each region, using the segmentation result from the initial scan as the reference. Region-wise brain volumes were computed as the total number of voxels in each label. To quantify inter-scanner variability, the coefficient of variation (CV) was computed for both DSC and volume for each brain region. Lower CV values indicate reduced variability and, thus, better harmonization consistency across scanners. Since FTHP contains only one subject, the CV values reported represent dispersion across different scanners for that single subject.

**Dataset #3: MASiVar** The Multisite, Multiscanner, and Multisubject Acquisitions for Studying Variability in Diffusion Weighted Magnetic Resonance Imaging (MASiVar) (Cai et al., 2021)[5] study provides a multi-subject, multi-scanner validation dataset. We utilized $T_1$-w MR images from Cohort II, comprising of five adult subjects, each scanned on three to four different scanners across three institutions. To focus on inter-scanner and inter-subject variability, repeat scans on the same scanner were excluded. For harmonization, the $T_1$-w

---

4. https://www.nitrc.org/projects/fthp

5. https://openneuro.org/datasets/ds003416/versions/2.0.2

image from each subject's initial scan at site #1 served as the harmonization target for both HACA3 and HACA3$^+$. Whole brain segmentation was performed using SLANT (Huo et al., 2019) for both unharmonized and harmonized images. For each subject, the CV for DSC and regional brain volume was calculated across their multi-site scans, with results reported as the mean and standard deviation over subjects. This approach allows assessment of harmonization consistency within subjects.

### 3.3. Ablation Study Using the ON-Harmony Dataset

The ON-Harmony (Warrington et al., 2025) dataset consists of T1-w and FLAIR MR images from 20 participants, each scanned on six 3T scanners across five sites and three major vendors. Repeat scans and additional available modalities were excluded from our experiments.

To evaluate the contribution of each component in HACA3$^+$, we conducted an ablation study using this dataset. Specifically, we evaluated four configurations: (i) HACA3, (ii) HACA3$^+$ without the artifact encoder, (iii) HACA3$^+$ without the large-scale training dataset, and (iv) HACA3$^+$ with all proposed enhancements. For all configurations, the harmonization target was defined as each subject's initial scanning session. Since the ON-Harmony dataset does not include limited FOV acquisitions, the foreground-aware attention mechanism was not explicitly ablated. In this setting, removing the enhanced attention mechanism would not affect performance, as no limited FOV conditions are present.

## 4. Experiments and Results

### 4.1. Region Imputation Using Enhanced Attention

Figure 1 shows qualitative examples for the limited FOV simulations using each image contrast from seven in-domain sites. The ground truth is the harmonized full FOV image. Quantitative results for the two types of limited FOV simulations using each image contrast are shown in Fig. 2. The PSNR and SSIM when using HACA3$^+$ are significantly higher for all image contrasts than when using HACA3. Statistical significance was computed using a paired Wilcoxon test with Bonferroni correction ($p$-value $< 0.0001$). Each image contrast was tested separately. For example, when the T$_1$-w image with simulated limited FOV was input, the other input source images included the full FOV. Qualitative examples of clinically acquired limited FOV images are shown in Fig. 3. These images do not have corresponding full FOV ground truth images and are only shown for qualitative purposes.

### 4.2. Inter-site Harmonization Using TREAT-MS Traveling Subjects

Figure 4 illustrates a representative healthy traveling subject from the TREAT-MS Traveling Subjects dataset. The input source images were obtained from a single non-Johns Hopkins imaging site and harmonized to the images acquired at the Johns Hopkins site. We quantitatively evaluate inter-site harmonization performance across the entire TREAT-MS Traveling Subjects dataset, reporting both PSNR and SSIM in Fig 5. Both HACA3 and HACA3$^+$ demonstrate comparable performance on this in-domain, full FOV dataset, with no statistically significant differences observed between the methods. This outcome indicates

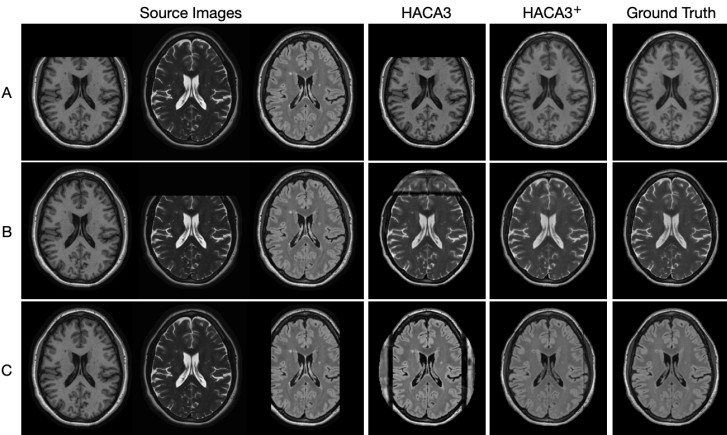

Figure 1: Qualitative results from the **(A)** $T_1$-w degraded anterior simulation, **(B)** $T_2$-w degraded anterior simulation, and **(C)** FLAIR degraded left/right simulation. Each row shows the different source images input to the model. We compare between HACA3 and HACA3$^+$. HACA3 uses the background mask from the first source image, which, in this scenario, is the $T_1$-w image. As a result, HACA3 does not attempt to impute in (A), but it does attempt to impute in (B) and (C).

that the methodological enhancements of HACA3$^+$ do not compromise harmonization quality in standard scenarios, thus preserving in-domain performance.

### 4.3. Inter-site Harmonization Using FTHP Dataset

Figure 6 presents the DSC variability across brain regions in the FTHP dataset. Harmonization leads to statistically significant improvement in DSC for eight out of nine regions compared to the unharmonized result. Significant differences between results were calculated using a paired Wilcoxon test with Benjamini–Hochberg correction. There were no statistically significant differences in DSC between the results using HACA3 and HACA3$^+$. Table 3 reports the CV for both DSC and volume, calculated across scans at different sites from the same subject. Since the FTHP dataset consists of a single subject scanned across multiple sites, only one CV value is provided per region. Harmonization consistently reduced CV for both DSC and volume in all regions. HACA3$^+$ and HACA3 outperformed the unharmonized result in terms of DSC and volume CV. The differences between HACA3$^+$ and HACA3 were not statistically significant.

### 4.4. Inter-site Harmonization Using MASiVar Dataset

Table 4 reports the CV for DSC and volume across brain regions in the MASiVar dataset. For each region, the CV is calculated across scans for each subject, and the values in the table represent the mean and standard deviation across the five subjects in Cohort II. Consistent with results in the FTHP dataset, harmonization reduced CV for both DSC and volume in all regions. HACA3 and HACA3$^+$ performed similarly, with only minimal differences

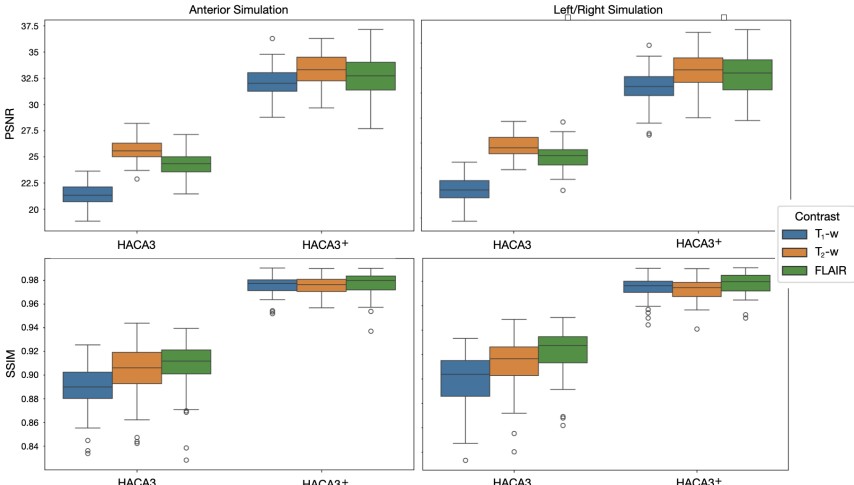

Figure 2: The PSNR and SSIM of the harmonized limited FOV images compared with the acquired full FOV image for each image contrast. We used two limited FOV simulations: anterior and left/right. Each contrast was tested separately. For example, when the $T_1$-w image had limited FOV, the $T_2$-w and FLAIR were full FOV. For each image contrast, HACA3$^+$ significantly ($p$-value $< 0.0001$) outperformed HACA3.

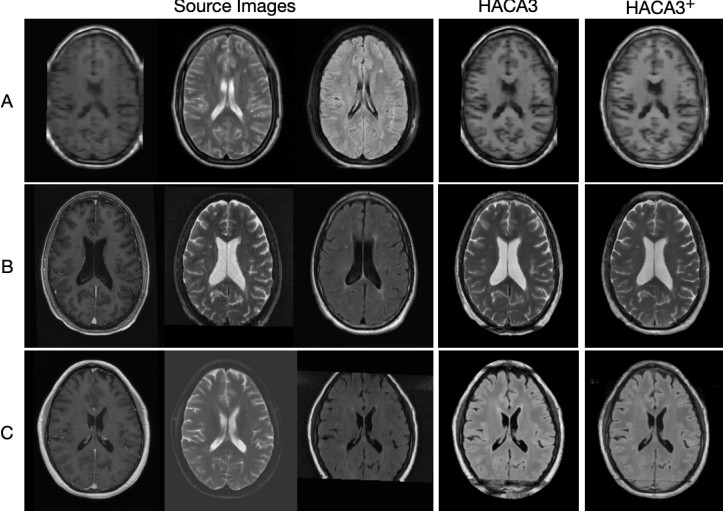

Figure 3: Qualitative results on real, clinically acquired MR images. Each row corresponds to a different person with MS. **(A)** shows the harmonized $T_1$-w result using a limited FOV $T_1$-w input image. **(B)** shows the harmonized $T_2$-w result using a limited FOV $T_2$-w input image. **(C)** shows the harmonized FLAIR result using a limited FOV FLAIR input image.

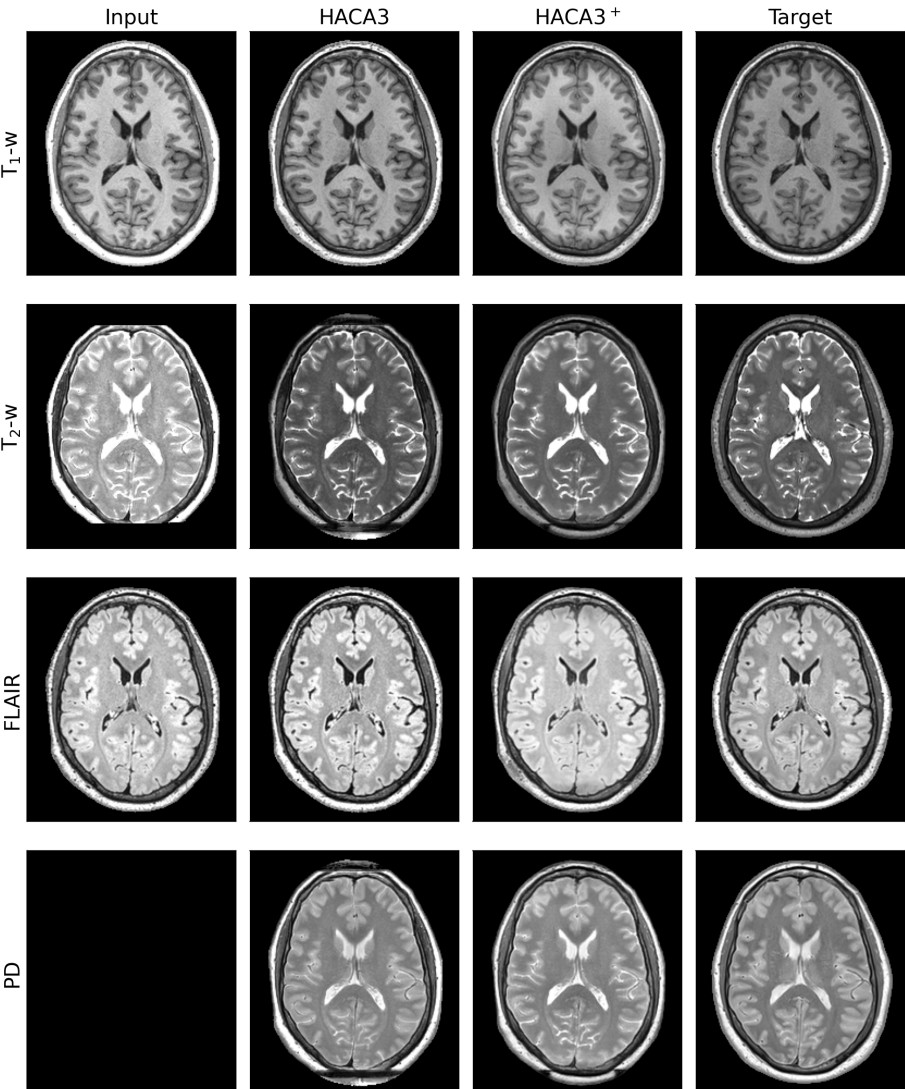

Figure 4: Inter-site harmonization results using HACA3 and HACA3$^+$. The input source images were acquired at a single non-Johns Hopkins site. The input PD was not included to demonstrate the imputation ability. The target images were acquired at the Johns Hopkins site.

between them. There were no statistically significant results in this dataset most likely due to a smaller number of subjects.

## 4.5. Ablation

Figure 7 illustrates representative harmonization results for an ON-Harmony subject across different configurations for T1w and FLAIR images, while Fig. 8 summarizes the corre-

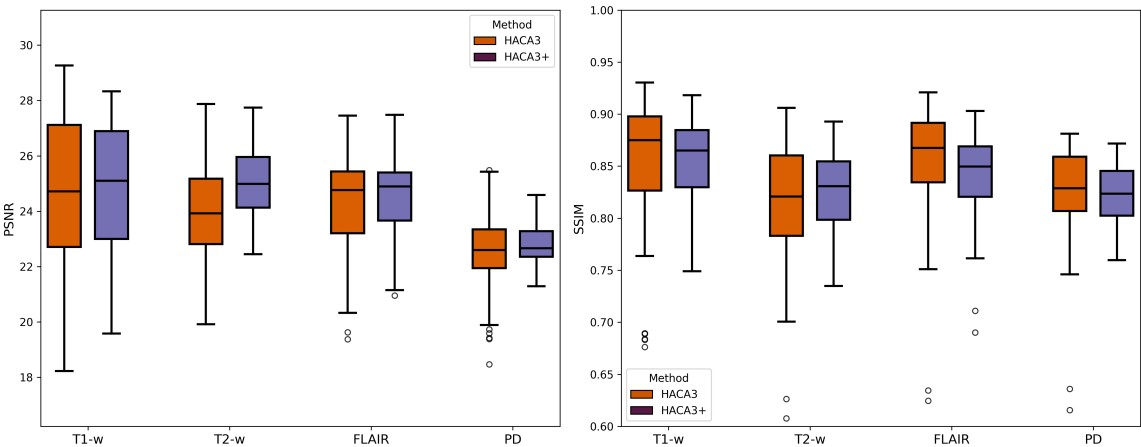

Figure 5: PSNR and SSIM of each harmonized MR image contrast over the TREAT-MS Traveling Subjects dataset. Acquired images were harmonized to the Johns Hopkins site. PSNR and SSIM were calculated using the acquired Johns Hopkins image as the reference image.

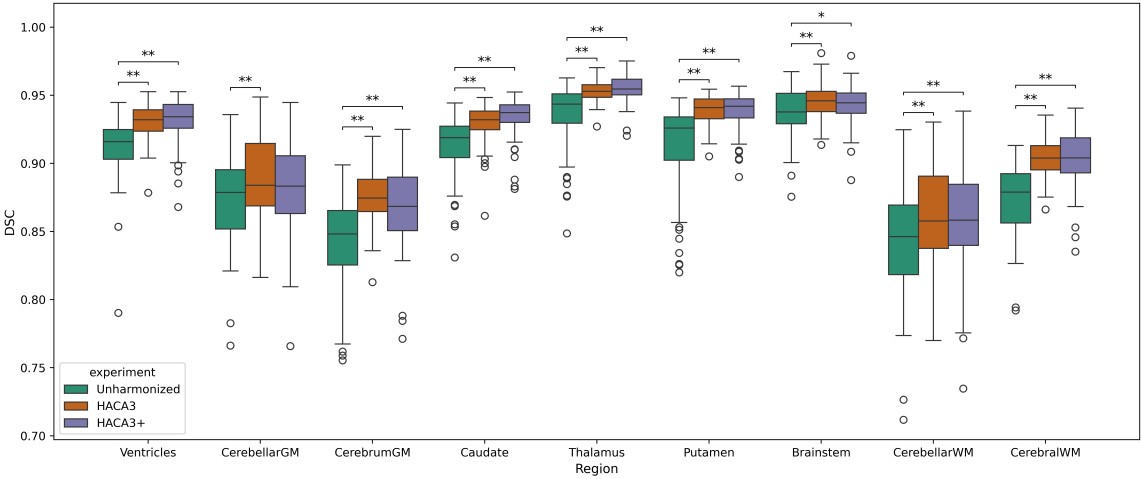

Figure 6: The DSC for each brain region computed using the segmentation result on the unharmonized, harmonized using HACA3, and harmonized using HACA3$^+$ T$_1$-w images. Significant differences between results are indicated using a paired Wilcoxon test with Benjamini–Hochberg correction (symbols indicate significance level (**: $p$-value $< 0.01$, *: $p$-value $< 0.05$)).

sponding PSNR and SSIM metrics. The full HACA3$^+$ model, incorporating all proposed enhancements, consistently outperforms both the ablated variants and the original HACA3. These findings suggest that the performance gains are primarily driven by the inclusion of the large-scale training dataset. Importantly, the ON-Harmony dataset was not used during

Table 3: The CV for DSC and volume (Vol) of each brain region computed using the segmentation result on the unharmonized, harmonized using HACA3, and harmonized using HACA3$^+$ T$_1$-w images from the FTHP dataset. The lowest DSC and Vol CV for each region are marked in **bold**.

| | Unharmonized | | HACA3 | | HACA3$^+$ | |
|---|---|---|---|---|---|---|
| **Region** | **DSC$_{CV}$** ↓ | **Vol$_{CV}$** ↓ | **DSC$_{CV}$** ↓ | **Vol$_{CV}$** ↓ | **DSC$_{CV}$** ↓ | **Vol$_{CV}$** ↓ |
| Ventricles | 0.0243 | 0.0408 | 0.0166 | 0.0292 | **0.0135** | **0.0262** |
| Cerebellar GM | 0.0382 | 0.0425 | 0.0379 | **0.0269** | **0.0346** | 0.0289 |
| Cerebrum GM | 0.0387 | 0.0498 | 0.0331 | **0.0161** | **0.0232** | 0.0217 |
| Caudate | 0.0234 | 0.0518 | 0.0153 | 0.0196 | **0.0147** | **0.0176** |
| Thalamus | 0.0231 | 0.0388 | 0.0097 | 0.0179 | **0.0075** | **0.0169** |
| Putamen | 0.0346 | 0.0452 | 0.0148 | 0.0192 | **0.0112** | **0.0191** |
| Brainstem | 0.0193 | 0.0402 | 0.0150 | **0.0120** | **0.0131** | 0.0217 |
| Cerebellar WM | 0.0461 | 0.0901 | 0.0444 | **0.0209** | **0.0438** | 0.0244 |
| Cerebral WM | 0.0289 | 0.0623 | 0.0221 | 0.0175 | **0.0164** | **0.0147** |

Table 4: The CV mean and standard deviation for DSC and volume (Vol) of each brain region computed using the segmentation result on the unharmonized, harmonized using HACA3, and harmonized using HACA3$^+$ T$_1$-w images for the 5 subjects in Cohort II of the MASiVar dataset. The lowest DSC and Vol CV for each region are marked in **bold**.

| | Unharmonized | | HACA3 | | HACA3$^+$ | |
|---|---|---|---|---|---|---|
| **Region** | **DSC$_{CV}$** ↓ | **Vol$_{CV}$** ↓ | **DSC$_{CV}$** ↓ | **Vol$_{CV}$** ↓ | **DSC$_{CV}$** ↓ | **Vol$_{CV}$** ↓ |
| Ventricles | 0.0405±0.0181 | 0.0401±0.0120 | **0.0283±0.0118** | 0.0408±0.0077 | 0.0287±0.0197 | **0.0259±0.0143** |
| Cerebellar GM | 0.0523±0.0170 | 0.0332±0.0114 | **0.0304±0.0201** | 0.0251±0.0125 | 0.0328±0.0135 | **0.0209±0.0042** |
| Cerebrum GM | 0.0409±0.0069 | 0.0662±0.0116 | 0.0262±0.0058 | 0.0231±0.0072 | **0.0258±0.0067** | **0.0154±0.0089** |
| Caudate | 0.0269±0.0070 | 0.0523±0.0122 | **0.0169±0.0065** | **0.0264±0.0094** | 0.0258±0.0088 | 0.0370±0.0104 |
| Thalamus | 0.0213±0.0053 | 0.0414±0.0080 | **0.0067±0.0022** | 0.0252±0.0126 | 0.0100±0.0050 | **0.0189±0.0062** |
| Putamen | 0.0239±0.0112 | 0.0340±0.0093 | **0.0147±0.0080** | **0.0223±0.0077** | 0.0213±0.0141 | 0.0228±0.0112 |
| Brainstem | 0.0323±0.0091 | 0.0275±0.0086 | 0.0220±0.0090 | **0.0173±0.0087** | **0.0180±0.0054** | 0.0178±0.0045 |
| Cerebellar WM | 0.0741±0.0226 | 0.0688±0.0249 | **0.0414±0.0254** | 0.0367±0.0161 | 0.0421±0.0223 | **0.0270±0.0098** |
| Cerebral WM | 0.0376±0.0084 | 0.0531±0.0248 | 0.0212±0.0054 | **0.0272±0.0087** | **0.0189±0.0054** | 0.0286±0.0077 |

training for either HACA3 or HACA3$^+$, underscoring the improved generalization capability of HACA3$^+$ to unseen multi-site data.

## 5. Discussion and Conclusion

In this validation paper, we performed comprehensive validation of the publicly available HACA3 MR harmonization model and an enhanced variant, HACA3$^+$. HACA3$^+$ incorporates three specific enhancements to address the limitations of HACA3. The first enhancement involved retraining the artifact encoder using more data and varying levels of simulated artifacts. The level of simulated artifacts was explicitly used as the margin in the triplet loss to develop a continuous space of low-to-high levels of artifacts. The second

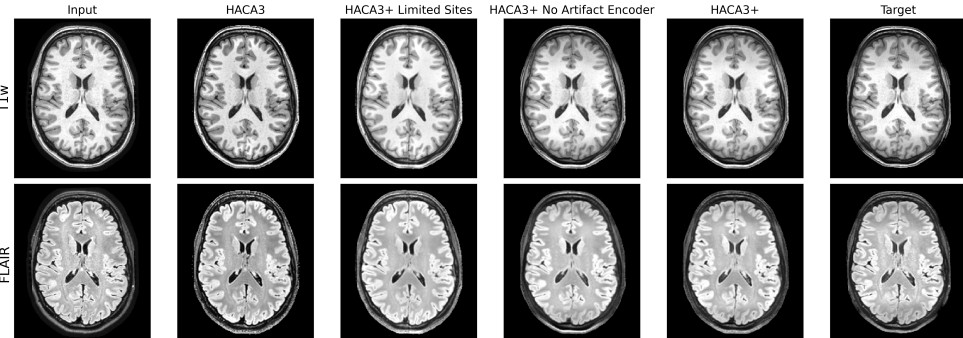

Figure 7: Representative harmonization results for an ON-Harmony subject using different model configurations for T1w and FLAIR images. The full HACA3$^+$ model produces outputs that are more consistent with the target domain contrast compared to ablated variants and the baseline HACA3.

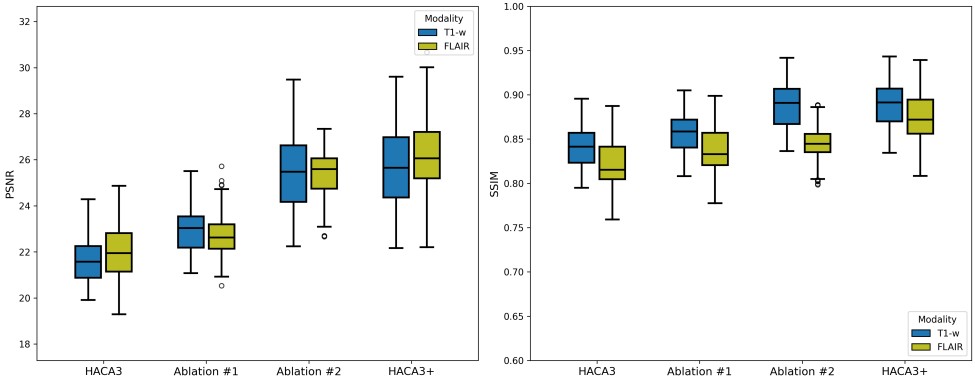

Figure 8: The PSNR and SSIM results from the ablation experiment using the ON-Harmony dataset. The harmonization target for each configuration was the subject's initial scanning session. Ablation #1 is HACA3 with only the attention and artifact enhancements. Ablation #2 is HACA3 with the larger training dataset, attention enhancement, and no artifact encoder.

enhancement introduced a spatially-aware attention mechanism to distinguish foreground and background, thereby improving local adaptation within each 2D slice of the source images. The third enhancement included training on a substantially larger amount of data, encompassing images from 132 scanners—to our knowledge, a scale unmatched by previous structural MR harmonization efforts.

On our private traveling subjects dataset (14 subjects within 9 sites), we found that both HACA3 and HACA3$^+$ exhibited strong, statistically indistinguishable performance in terms of PSNR and SSIM, showing that both models generalize well to in-domain, multi-site data.

On the FTHP dataset involving a single subject who traveled to 116 different sites, harmonization with either model led to significant reductions in inter-scanner variability, measured as the CV for both DSC and regional brain volume using SLANT labels. Notably, HACA3$^+$ achieved the lowest CV across most brain regions and minor DSC improvements in the ventricles and deep gray matter structures such as the caudate and thalamus. Although direct comparisons between HACA3 and HACA3$^+$ were not statistically significant. We note that ground truth segmentation were not available, so comparisons relied on the reference segmentation from the initial scan.

Analysis on the MASiVar dataset involving five subjects scanned at multiple sites further confirmed these findings. Harmonization led to reduced CV for both DSC and regional brain volumes with each subject across the scanners, although the limited sample size ($N = 5$) precluded statistical significance between methods. Our study demonstrates that HACA3$^+$'s methodological improvements do not compromise performance on standard full FOV datasets, maintaining strong harmonization reliability. Critically, the attention mechanism enhancement of HACA3$^+$ benefits in scenarios involving limited FOV or regional dropout, an area where HACA3 had limitations. Importantly, the model is conservative in its predictions. Regions absent in all source images are not imputed, which mitigates the risk of hallucination.

The third enhancement did not directly lead to statistically significant improvement in HACA3. We trained HACA3$^+$ on $6\times$ more scanners than HACA3 and more than $4\times$ the number of subjects. Although, we also did not modify the HACA3 architecture. Due to the substantial increase in training data, it is possible that the model architecture needs to be adapted to handle more complexity, which could be a reason why we did not observe a statistically significant improvement. Our experiments were geared towards validation of harmonization on traveling subject datasets and not on methodological development of a harmonization algorithm. This is an area for future exploration.

Our validation focuses primarily on healthy subjects and individuals with MS, reflecting the TREAT-MS pragmatic clinical trial dataset that contributed the bulk of our training data. The models' performance in other populations has not been tested. Furthermore, harmonization is currently limited by the artifact encoder's ability to detect and mitigate poor quality images. Extension to diverse disease populations and architecture development are promising directions of future work.

A limitation of the artifact encoder involves it being trained on simulated single-artifact cases across a spectrum of severity. We acknowledge that real-world acquisitions often present with multiple, overlapping artifact types, which may not be fully captured by the encoder. Robustness to unseen or extreme artifact combinations remains an open challenge. At an architectural level, our current approach relies on 2D, slice-wise attention mechanisms. Given spatially complex artifacts and pathologies, we hypothesize that a 3D harmonization and attention strategy could significantly improve robustness and anatomical fidelity. Our ablation experiment on an out-of-domain dataset demonstrates the impact of expanding the training dataset. In summary, HACA3$^+$ establishes a new benchmark in multi-site harmonization, trained on a large and diverse dataset, and rigorously validated on multiple traveling subject datasets. The model's enhancements improve adaptability to real-world clinical variability without sacrificing in-domain reliability. The code and pre-trained weights are made publicly available to facilitate future research and clinical translation.

## Acknowledgments

This research is partially supported by the Johns Hopkins University Percy Pierre Fellowship (Hays) and the National Science Foundation Graduate Research Fellowship under Grant No. DGE-2139757 (Hays). Development is partially supported by FG-2008-36966 (Dewey), CDMRP W81XWH2010912 (Prince), NIH R01EB036013 (Prince), NIH R01 CA253923 (Landman), NIH R01 CA275015 (Landman), the National MS Society grant RG-1507-05243 (Pham) and Patient-Centered Outcomes Research Institute (PCORI) grant MS-1610-37115 (Newsome and Mowry). The statements in this publication are solely the responsibility of the authors and do not necessarily represent the views of the Patient-Centered Outcomes Research Institute (PCORI), its Board of Governors or Methodology Committee.

This research was supported in part by the Intramural Research Program of the National Institutes of Health (NIH). The contributions of the NIH author(s) were made as part of their official duties as NIH federal employees, are in compliance with agency policy requirements, and are considered Works of the United States Government. However, the findings and conclusions presented in this paper are those of the author(s) and do not necessarily reflect the views of the NIH or the U.S. Department of Health and Human Services.

Data were provided [in part] by the Human Connectome Project, WU-Minn Consortium (Principal Investigators: David Van Essen and Kamil Ugurbil; 1U54MH091657) funded by the 16 NIH Institutes and Centers that support the NIH Blueprint for Neuroscience Research; and by the McDonnell Center for Systems Neuroscience at Washington University.

Data were provided [in part] by OASIS-3: Longitudinal Multimodal Neuroimaging: Principal Investigators: T. Benzinger, D. Marcus, J. Morris; NIH P30 AG066444, P50 AG00561, P30 NS09857781, P01 AG026276, P01 AG003991, R01 AG043434, UL1 TR000448, R01 EB009352.

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
