# OpenReview forum: "Harmonizing MR Images Across 100+ Scanners: Multi-site Validation with Traveling Subjects and Real-world Protocols"
_MIDL.io/2026/Validation_Papers — MIDL 2026 - Validation Papers Poster_

### Official Review · Reviewer_HCD6 · 2025-12-17

**Confidence:** 3
**Preliminary Rating:** 5
**Final Rating:** 5

**Summary:**

This paper presents HACA3+, an enhanced version of the HACA3 MRI harmonization framework, designed for large-scale, multi-site harmonization across more than 100 scanners. The authors introduce three targeted improvements—an artifact-aware encoder, foreground/background-sensitive attention, and substantially expanded multi-site training—and validate the method using multiple traveling-subject datasets and real-world acquisition protocols. Results show that HACA3+ maintains robust in-domain harmonization performance while improving reliability in challenging scenarios such as limited field-of-view acquisitions. The study emphasizes large-scale validation and clinical realism over architectural novelty.

**Strengths:**

1. Exceptionally large and diverse training corpus. Training across 64 sites and 132 scanners represents an unprecedented scale for image-based MRI harmonization and substantially exceeds prior work, directly addressing generalization concerns common in multi-site studies.
2. Validation on three independent traveling-subject datasets is a major strength, as such datasets are widely regarded as the gold standard for assessing inter-scanner harmonization robustness.
3. The inclusion of limited field-of-view simulations and real-world truncated acquisitions reflects practical clinical challenges often ignored in harmonization studies.
4. The model explicitly avoids hallucinating anatomy when regions are absent in all source images, which is an important safety consideration for clinical deployment.

**Weaknesses:**

I do not see any major weaknesses in this article, and only two minor issues merit mention:
1. Validation relies primarily on image similarity metrics and whole-brain segmentation consistency using a single segmentation method. Maybe other clinically relevant downstream tasks (e.g., lesion quantification, cortical thickness estimation)  could be explored.
2. The improved artifact encoder is motivated clearly, but its standalone effectiveness is not explicitly evaluated (e.g., artifact detection accuracy or correlation with human quality ratings).

**Detailed Comments:**

1. Consider including at least one additional downstream analysis (e.g., lesion volume stability or cortical thickness consistency) to strengthen translational relevance.
2. Consider reporting confidence intervals alongside CV and DSC metrics to improve the statistical clarity.
3. Clarify whether segmentation results depend on harmonized intensity distributions or are influenced by residual scanner-specific artifacts.

**Justification Of Final Rating:**

This paper delivers a rigorous, large-scale, and clinically grounded evaluation of MRI harmonization. Although the methodological innovations are incremental and performance gains over HACA3 are sometimes modest, the scale of training, use of traveling-subject validation, and emphasis on real-world robustness provide substantial value to the community.

**Justification Of The Preliminary Rating:**

This paper delivers a rigorous, large-scale, and clinically grounded evaluation of MRI harmonization. Although the methodological innovations are incremental and performance gains over HACA3 are sometimes modest, the scale of training, use of traveling-subject validation, and emphasis on real-world robustness provide substantial value to the community.

**Questions To Address In The Rebuttal:**

1. How much of the observed improvement in HACA3+ can be attributed specifically to architectural modifications versus increased training data?
2. How robust is the artifact encoder’s severity scoring when applied to unseen artifact types or extreme acquisition failures?
3. Would a fully 3D training strategy materially change harmonization performance, particularly for spatially complex artifacts?

---

### Official Review · Reviewer_9hJT · 2026-01-09

**Confidence:** 4
**Preliminary Rating:** 2

**Summary:**

This paper proposes HACA3+, an upgraded version of HACA, introducing two main modifications to the Artifact Encoder architecture and the attention mechanism used in HACA. These updates lead to consistent performance improvements. The authors conduct extensive experiments using data collected from dozens of institutions, as well as multiple travelling subject datasets, which aligns well with the theme of the Validation Track. However, the manuscript suffers from poor writing clarity, and the experimental section lacks baseline comparisons beyond HACA itself, which significantly weakens the persuasiveness of the work.

**Strengths:**

1. The paper builds upon HACA and introduces effective upgrades to both the Artifact Encoder structure and the attention mechanism.

2. Experiments are conducted on data from dozens of institutions and multiple travelling subject datasets, which is highly relevant to the Validation Track.

3. The experimental design is comprehensive, and the results are strong and convincing within the presented setting.

**Weaknesses:**

1. The writing quality is a major concern. The paper barely describes the main architecture of HACA, and instead directly focuses on the two proposed modifications. This makes the paper difficult to follow. In general, the authors should first clearly explain the shared or previously proposed base architecture.

2. In the experimental section, the authors only compare HACA+ with HACA, without including any other baselines. Given that HACA was published in 2023, the authors should at least include comparisons with two to three more recent or competitive baseline methods.

3. The analysis and discussion of related work are insufficient.

**Detailed Comments:**

Refer to the weaknesses.

**Justification Of The Preliminary Rating:**

I think the experimental setup is thorough, the dataset scale is very large, and the use of multiple travelling subject datasets is a strong aspect of this work. However, the paper’s presentation is poor, as it does not clearly describe the overall architecture of HACA+, focusing only on the differences from HACA. Moreover, the lack of baselines beyond HACA makes it difficult to assess whether HACA+ truly represents a state-of-the-art method.

**Questions To Address In The Rebuttal:**

Refer to the weaknesses.

---

### Official Review · Reviewer_ybY9 · 2026-01-10

**Confidence:** 5
**Preliminary Rating:** 4
**Final Rating:** 4

**Summary:**

The paper extends the HACA3 model with architectural enhancements (spatially-aware attention mechanism) and a significantly larger training dataset. These improvements allow HACA3+ to better handle real-world clinical variability and perform anatomy imputation for limited FOV images, while maintaining the strong in-domain performance established by HACA3.

**Strengths:**

Similiar to HACA3, the experiments in this paper are well designed. A large amount of data was collected to avoid bias. The artifact encoder was re-trained with more datasets. Figure 2 shows the promising improvement of the newly designed framework when it comes to limited FOV images. The paper also compared HACA3 and HACA3+ using incomplete source images. It showed that HACA3+ has the ability to generate full field-of-view (FOV) results even when the input images have a limited FOV.

**Weaknesses:**

In Figure 6, HACA3+ yields lower DSC scores than HACA3 in certain regions, such as the Cerebrum GM. Furthermore, Table 4 indicates that HACA3 outperforms HACA3+ in terms of DSC and Volume CV for the majority of regions. While Table 3 demonstrates the advantages of HACA3+ across most areas, these results may be biased due to the limited dataset size (single subject). No ablation study is shown in the paper to isolate and quantify the individual contributions of each of the three proposed enhancements: 1.the improved artifact encoder, 2.the background-sensitive attention mechanism, 3.the expanded training dataset.

**Detailed Comments:**

The quality of the figures is inconsistent. Specifically, figure 2 and 6 are in high resolution, while figure 5 is in lower quality.

**Justification Of Final Rating:**

The paper is a meaningful contribution overall, but the key ablation is still missing. Since the rebuttal only promises an ablation of the three enhancements for the camera-ready version, I’m keeping my rating at 4 (Weak Accept), and I’m looking forward to seeing the full ablation results in the future version.

**Justification Of The Preliminary Rating:**

While Table 4 indicates that HACA3+ yields slightly lower DSC scores in certain area, the model demonstrates a critical advantage in generating robust results from limited Field-of-View (FOV) inputs. This capability is essential for practical clinical implementation, as full FOV acquisitions are not always available.

**Questions To Address In The Rebuttal:**

Do you have ablation results that isolate the individual contributions of the three specific enhancements (artifact encoder, attention mechanism, and dataset scale) to determine which is responsible for the performance gains in limited FOV tasks versus the slight degradation in standard segmentation metrics?

---

### Author Rebuttal · Authors · 2026-01-25

We appreciate the reviewers’ valuable and constructive suggestions. Below, we address major concerns and outline our planned revisions accordingly. More details are added as official comments to each reviewer. Planned changes to the paper in response to reviewer comments are marked with an asterisk * in this rebuttal and will be updated upon acceptance for the camera-ready manuscript.

*R1/ybY9: Figure 5 will be regenerated at a higher resolution.

*R1/ybY9 and R3/HCD6: We are currently working on a full ablation study using the ON-Harmony dataset to isolate the individual contributions of the three specific enhancements to be included in the camera-ready version. We were not able to complete this ablation within the rebuttal period and thus cannot include results in this rebuttal..

*R2/9hJT: To improve on the writing quality, we will add more background and description of HACA3’s architecture in the Introduction.

*R2/9hJT: We will cite additional recent related work in the Introduction and provide more motivation for our Validation paper.

*R3/HCD6: We will add additional discussion points based on the interesting questions and comments mentioned by this reviewer involving limitations, a 3D version, and the artifact encoder’s robustness.

---

### Meta-Review · Area_Chair_GjAR · 2026-02-03

**Recommendation:** Accept (Poster)
**Confidence:** 4

**Metareview:**

This paper presents HACA3+, an enhanced MRI harmonization framework trained on data from 100+ scanners and validated using multiple traveling-subject datasets and real-world protocols. Reviewers agree the primary contribution is the unusually rigorous, clinically realistic, large-scale validation (including limited-FOV scenarios), which is well aligned with the Validation Track and demonstrates strong robustness and practical utility.

Key concerns are (i) the lack of an ablation study to disentangle the effects of the artifact encoder, attention mechanism, and expanded training data; (ii) limited baselines beyond HACA3; and (iii) writing clarity (insufficient description of the base HACA3/HACA3+ architecture) plus minor figure quality issues. The authors’ rebuttal adequately addresses these points by committing to a full ablation in the camera-ready, improving architectural exposition and related work, and regenerating low-resolution figures, with additional discussion on limitations (artifact robustness) and future directions (e.g., 3D variants, broader downstream tasks). Overall, despite deferred analysis, the work’s validation depth and realism outweigh presentation gaps, warranting acceptance.

---

### Decision · Program_Chairs · 2026-02-14

Accept (Poster)